# Evolution of phenotypic fluctuation under host-parasite interactions

**Naoto Nishiura**[1], **Kunihiko Kaneko**[1,2]*

**1** Department of Basic Science, Graduate School of Arts and Sciences, University of Tokyo, Tokyo, Japan,
**2** Center for Complex Systems Biology, Universal Biology Institute, University of Tokyo, Tokyo, Japan

* kaneko@complex.c.u-tokyo.ac.jp

**Data Availability Statement:** All relevant data are within the manuscript and its Supporting information files. Code developed as a part of this study are available at:https://github.com/NaotoNishiura/Evolution-of-Phenotypic-Fluctuation.git.

## Abstract

Robustness and plasticity are essential features that allow biological systems to cope with complex and variable environments. In a constant environment, robustness, i.e., insensitivity of phenotypes, is expected to increase, whereas plasticity, i.e., the changeability of phenotypes, tends to diminish. Under a variable environment, existence of plasticity will be relevant. The robustness and plasticity, on the other hand, are related to phenotypic variances. As phenotypic variances decrease with the increase in robustness to perturbations, they are expected to decrease through the evolution. However, in nature, phenotypic fluctuation is preserved to a certain degree. One possible cause for this is environmental variation, where one of the most important "environmental" factors will be inter-species interactions. As a first step toward investigating phenotypic fluctuation in response to an inter-species interaction, we present the study of a simple two-species system that comprises hosts and parasites. Hosts are expected to evolve to achieve a phenotype that optimizes fitness. Then, the robustness of the corresponding phenotype will be increased by reducing phenotypic fluctuations. Conversely, plasticity tends to evolve to avoid certain phenotypes that are attacked by parasites. By using a dynamic model of gene expression for the host, we investigate the evolution of the genotype-phenotype map and of phenotypic variances. If the host–parasite interaction is weak, the fittest phenotype of the host evolves to reduce phenotypic variances. In contrast, if there exists a sufficient degree of interaction, the phenotypic variances of hosts increase to escape parasite attacks. For the latter case, we found two strategies: if the noise in the stochastic gene expression is below a certain threshold, the phenotypic variance increases via genetic diversification, whereas above this threshold, it is increased mediated by noise-induced phenotypic fluctuation. We examine how the increase in the phenotypic variances caused by parasite interactions influences the growth rate of a single host, and observed a trade-off between the two. Our results help elucidate the roles played by noise and genetic mutations in the evolution of phenotypic fluctuation and robustness in response to host–parasite interactions.

**Funding:** This work was supported by grants 17H06386 (Grant-in-Aid for Scientific Research on Innovative Areas) and 20H00123 (Grant-in-Aid for Scientific Research (A)) from the The Ministry of Education, Culture, Sports, Science and Technology (MEXT) of Japan (https://www.mext.go.jp/en/index.htm) to K.K. The funders had no role in study design, data collection and analysis, decision to publish, or preparation of the manuscript.

**Competing interests:** The authors have declared that no competing interests exist.

## Author summary

Plasticity and phenotypic variability induced by internal or external perturbations are common features of biological systems. However, under evolution for given environmental conditions, phenotypic variability is not advantageous, because it leads to the deviation from the fittest state. This has been demonstrated by previous laboratory and computer experiments. As a possible origin for the remnant phenotypic variance, we investigated the role of host–parasite interactions such as those between bacteria and phages. Different parasite-types attack hosts of certain phenotypes. Through numerical simulations of the evolution of the host genotype–phenotype mapping, we found that hosts increase phenotypic variation by increasing phenotypic fluctuations if the interaction is sufficiently strong. Depending on the degree of noise in gene expression dynamics, there are two distinct strategies for increasing phenotypic variances: stochasticity in gene expression or genetic variances. The former strategy, which can work over a faster time scale, leads to a decline in fitness, whereas the latter reduces the robustness of the fitted state. Our results provide insights into how phenotypic variances are preserved and how hosts can escape being attacked by parasites whose genes mutate to adapt to changes in parasites. These two host strategies, which depend on internal and external conditions, can be verified experimentally via the transcriptome analysis of microorganisms.

## Introduction

Robustness and plasticity are two important properties of biological systems. To maintain function and high fitness against internal noise, environmental variation, and genetic changes, the fitted state must be robust to such disturbance, whereas the phenotype needs to possess plasticity to adapt to environmental variation. Indeed, the evolution of robustness and plasticity has been investigated extensively both theoretically and experimentally [1–5].

In this context, phenotypes are shaped by dynamic processes involving several genetically determined variables, as well as external and internal noise. Hence phenotypes can vary not only as a consequence of genetic variation, but even between isogenic individuals [6–10]. On the other hand, the dynamical process varies by the genetic changes, so that there also exists phenotypic fluctuations by the genetic variation [11–13]. The sum of both sources of phenotypic variation defines the genotype–phenotype mapping [14–16]. The evolution of genotype–phenotype mapping, thus, is essential to understand the evolution of robustness and plasticity, and has been explored extensively both in numerical [4, 17] and laboratory experiments [18, 19].

The magnitude of these phenotypic fluctuations generally decreases as a result of evolution under a given environment, because, with this decrease, the system deviates less from the fitted state [20–23]. In fact, experiments and simulations of adaptive evolution under fixed conditions have shown that fluctuations decrease over the course of evolution [24], and evolvability, i.e., the rate of increase in the fitness or the phenotypic change per generation, declines. Accordingly, as evolution progresses, the robustness of the phenotype increases, while the phenotypic fluctuations decrease. The decreased phenotypic fluctuation will then be accompanied by the decreased adaptability to environmental variation, as discussed by generalizing the fluctuation-response relationship in statistical physics [17, 25, 26].

In nature, however, phenotypic fluctuations and evolutionary potentiality persist. Even after evolution in a constant environment, the phenotype in question is not necessarily concentrated on its optimal value, instead its variance often keeps a rather large value. For

instance, even for isogenic cells (clones), fluctuations in the concentration of each protein remain sufficiently high. This leads to the following questions: Why are phenotypic fluctuations not reduced further, which, in principle, would be possible by evolving appropriately negative feedback processes for stabilization? How are phenotypic fluctuations or plasticity sustained?

One possible cause for the preservation of plasticity or fluctuation is environmental fluctuation [27–29]. In natural populations, environment is not fixed; instead, it is variable. If phenotypic fluctuation is reduced excessively, the plasticity required to adapt to new environmental conditions would be lost. The plasticity of a biological system, would then decrease its capacity to cope with environmental changes.

For every species, interactions with other species are one of the most important "environmental" factors that affect its existence. Even if a certain "external" environmental condition itself is fixed, the types and populations of other species may change because of species–species interactions [30]. For example, in the interaction between hosts (prey) and parasites (predators), the former may change their phenotype to escape the attack by latter's, which, in turn, will change the distribution of the latter phenotypes to continue attacking the former [31, 32]. Thus, each species may retain evolutionary plasticity to cope with dynamic changes in other species. Furthermore, phenotypic variation in one species may influence the other species, which can result in a dynamic change in inter-species interactions. If sufficient species–species interaction exists, the dynamic variation of phenotypes may be mutually sustained across multiple species.

As a first step to investigate the phenotypic fluctuations originating from such species–species interactions, we present a study of a simple two-species system that comprises hosts and parasites. Within this system, distinct parasite types that can attack hosts by possessing a specific phenotype exist; the host phenotypes are determined by genotype–phenotype mapping that incorporates possible fluctuations. Hosts are expected to evolve to achieve a phenotype that optimizes fitness and increases the robustness of the successful phenotype by reducing phenotypic fluctuations. in addition, they are expected to evolve plasticity, i.e., phenotypic adaptability, to cope with parasite attacks. We use a host gene-expression dynamics model to study the evolution of genotype–phenotype mapping and the resultant phenotypic fluctuations, and explore the evolution of phenotypic robustness and plasticity in response to interactions with parasites.

Here, we address the following questions: first, how are robustness and plasticity of phenotypes, which tend to exhibit opposite trends, maintained through evolution caused by interactions between the host and the parasite species? Second, are phenotypic fluctuations sustained to cope with parasite attacks? As mentioned before, there are two sources of phenotypic fluctuations: genetic variation and stochasticity in genotype–phenotype mapping (i.e., noise in the dynamics that shape the phenotype). Then, we address a third question: which of these sources of phenotypic fluctuation is dominant in sustaining the level of phenotypic fluctuation? Finally, we discuss the dependence of the evolution of phenotypic fluctuation and plasticity on the strength of the host–parasite interaction and the sensitivity of phenotype dynamics to noise.

## Models

### Gene regulatory networks

We employed a simple model for gene expression dynamics based on a sigmoidal function to study the evolution of the genotype–phenotype map. There are dynamics of protein expression level for each protein (gene) $i$. This model comprises a network consisting of genes that

mutually activate or inhibit each other's expression. In this model, there are $M$ genes whose gene expression level, $x_i^{(l)}$ ($i = 1, 2, \ldots, M$), corresponding to the host genotype $l$ at time $t$ is described by

$$\frac{dx_i^{(l)}(t)}{dt} = \gamma \left( f(\sum_{j>k}^{M} \frac{J_{ij} x_j^{(l)}(t)}{\sqrt{\alpha M}} + I_i - \theta_i) - x_i^{(l)}(t) + \epsilon \right) + \sigma \eta_i^{(l)}(t), \tag{1}$$

where $J_{ij}$ denotes the matrix representing the influence of gene $j$ on the expression of gene $i$. When $j$ activates (represses) the expression $i$, it takes the form $J_{ij} = 1(-1)$, while $J_{ij} = 0$ if $j$ does not influence $i$. The sigmoidal function $f(z)$ is given by

$$f(z) = \frac{1}{1 + \exp(-\beta(z))}, \tag{2}$$

with $\beta = 25$; this implies that $f(z)$ closely resembles a step function. Furthermore, $\theta_i$ represents the threshold of the input required for the expression of gene $i$. Therefore, whether the gene (protein) is expressed ($f(z) \approx 1$) or not ($f(z) \approx 0$) depends on whether the sum of inputs from the expression of other genes(proteins) is larger than $\theta_i$. In Eq (1), the value $\theta_i$ is selected randomly from a uniform distribution between 0.05 and 0.3. The term representing the interaction with other genes is scaled by $\sqrt{\alpha M}$, where $\alpha$ denotes the path density, that is, the fraction of nonzero values of $J_{ij}$. We adopted this scaling so that $x_i$ is comparable with the order of $\theta_i$ regardless of total gene number $M(= 64)$. (Because the connection is sparse, many of $J_{ij} = 0$, and the average number of inputs for gene is $\alpha M$. We normalized the interaction with other genes by the square root of the path number $\alpha M$ so that the variance of $J_{ij}$ is independent of $\alpha M$.) Moreover, there are $l_{\text{out}}(= 8)$ "output genes" $i = 1, 2, \ldots, l_{\text{out}}$, which determine the fitness as defined below. To eliminate the potential for the output genes to influence others, the summation is taken only for $j > l_{out}$. The term $\eta^{(l)}(t)$ denotes a Gaussian white noise originating from molecular fluctuations in chemical reactions.

Here, the noise is applied to the expression of each gene, and it is not correlated across genes. The value of $\sigma$ represents the strength of this noise (In the case where the gene expression of $x$ is negative, we set $x$ to zero.) For constant $f(z)$, Eq (1) defines an Ornstein-Uhlenbeck process, and to give the equilibrium variance of $\sigma^2/(2\gamma)$. Initially, all gene expression levels are set smaller than the threshold $\theta_i$. Furthermore, $\epsilon$ denotes the spontaneous expression level, which is smaller than $\theta_i$. $I_j$ denotes the external input for $l_{\text{inp}}(= 8)$ "input genes," with $I_j = 1$ for $j = M - l_{\text{inp}} + 1, M - l_{\text{inp}} + 2, \ldots, M$, and $I_j = 0$ for $j \leq M - l_{\text{inp}}$. The input genes are always/constitutively expressed. In order for $x_i$ to be expressed, $x_i$ must receive external or internal inputs from other $x_j$ values to advance beyond $\theta_i$.

Initially, we select $J_{ij}$ at random to obtain $J_{ij} = 1(-1)$ with a probability of 0.15 so that $\alpha = 0.3$. We set the initial number of host cells as $N$ and total number is constant at $N = 300$ throughout all simulations. Here, we set $M = 64$, $l_{\text{out}} = 8$, and $l_{\text{inp}} = 8$. The model parameters used in this study are set as in Table 1.

## Raw fitness and reproduction

The raw fitness of the host alone, i.e., in the absence of parasites, is determined by the fraction of output genes that are expressed as

$$\mu^{(l)}(t) = \mu_0 \sum_{j=1}^{l_{\text{out}}} x_j^{(l)}(t) \tag{3}$$

where $x_j$ denote the expression of the output genes and $\mu_0$ denotes the basal fitness. Therefore,

**Table 1. Model parameters and variable definitions.**

| Symbol | Description | value |
|---|---|---|
| $M$ | Total number of host genes | 64 |
| $N$ | Total number of host cells | 300 |
| $l_{\text{inp}}$ | Number of input genes | 8 |
| $l_{\text{out}}$ | Number of output genes | 8 |
| $I_j$ | Input strength | 1 |
| $\gamma$ | Time constant | 0.1 |
| $\alpha$ | Average path density | 0.3 |
| $\epsilon$ | Spontaneous expression level | 0.01 |
| $\beta$ | Sensitivity in the expression | 25 |
| $\mu_0$ | Basal growth rate | 0.0002 |
| $m$ | Mutation rate | 0.03 |
| $l_p$ | Total number of parasite genes | 3 |
| $N_p$ | Number of parasite types | 8 |
| $T_p$ | Parasite generation time | 100 |
| $D_p$ | Mutation rate of parasite | 0.01 |

the state with all output genes expressed ($x_j = 1$, for $j = 1, 2, \ldots, l_{\text{out}}$) is an optimal phenotype for achieving maximal growth.

We introduce parasite attacks into the model. There are $N_p = 2^{l_p}$ parasite genotypes that are coded as $i_1, i_2, \ldots, i_{l_p}$ with $i_j = 0$ or 1. For example, for $l_p = 3$, there are $2^3 = 8$ genotypes coded by 000, 001, ..., 111. Each parasite attacks hosts that possess the corresponding target gene expression, $i_1, i_2, \ldots, i_{l_p}$, that is, each parasite attacks the host whose gene expression pattern, $x_j$ ($1 \leq j \leq l_p$), matches ($i_1, \ldots, i_{l_p}$). Here, we term the $l_p$ genes as "target genes." The target condition for the attack of each parasite is given by $s(x_i) = 1$ for $x_i > 0.5$ and $s(x_i) = 0$ otherwise (as the threshold function is close to a step function, $x_i$ takes either $\sim 0$ or $\sim 1$). For instance, a host whose expression pattern $(s(x_1), s(x_2), s(x_3)) \approx (0, 1, 0)$ is attacked by the parasite ($i_1, i_2, i_3) = (0, 1, 0)$. Let $m$ ($m = 1, \ldots, 2^{l_p}$) be the parasite type whose $i_1, i_2, \ldots, i_{l_p}$ matches with host $x_i^{(l)}$ for $i = 1, \ldots, l_p$. The growth rate $\hat{\mu}_i^{(l)}$ of each host genotype $l$ decreases owing to the effect of the parasite $m$ as

$$\hat{\mu}^{(l)}(t) = \mu_0 \left( \sum_{j=1}^{l_{\text{out}}} x_j^{(l)}(t) - cP_m \right) \tag{4}$$

here $P_m$ denotes the population density of the m-th parasite that matches with the host gene expression $x_i^{(l)}$ for ($i = 1, \ldots, l_p$) as mentioned and $c$ denotes the strength coefficient of the attack by the parasite. The evolution of hosts is modelled by an individual-based approach with a fixed population size akin of the Moran process. Each host has its own genotype (gene regulatory networks) and the growth rate $\hat{\mu}$ determined by the gene expression levels calculated by numerical simulations in Eq (1). The volume of the host cells grows according to

$$\frac{dv(t)^{(l)}}{dt} = \hat{\mu}^{(l)}(t)v(t)^{(l)} \tag{5}$$

When the volume exceeds a threshold value of 2 ($v(0)$ set to 1), the host cell is divided into two parts. The total number of hosts $N$ is set to be constant over time, and thus, whenever a cell divides, another cell is randomly selected and removed from the original population.

Here, after division, the initial expression of $x_i$ is reset to take a random value smaller than the threshold $\theta_i$. The division time of the host is determined by the individual growth rate that ranges between 500 and 1,000 time units. At this time scale, the host gene expression reaches a steady state. The gene expression of hosts reaches a steady state after around 200 time units. Mutation is introduced in the division process and added to the network $J_{ij}$ with a low mutation rate $m = 0.03$. An $(i, j)$ path in the network matrix $J_{ij}$ is selected at random and its value is changed to one of the others to preserve the total path number. For example, if $J_{ij} = 1(−1)$ and $J_{kj} = 0$, then they are changed to $J_{kj} = 1(−1)$ and $J_{ij} = 0$. Here, we randomly select a gene $k \neq i$, $i \neq j, k \neq j$. Then, the path $j \to i$ is replaced by $j \to k$ to represent the mutation of a path in gene regulatory networks. We prohibited direct connections between the "input" and "output" genes. Initially, $J_{ij}$ is selected randomly with ±1 with the rate $\alpha = 0.3$, and $\alpha$ remains constant throughout the simulation. (However, the result is independent of this choice.)

It should be noted that once cells settle down to different fixed-point attractors, no more switching occurs in time. The on/off difference in the gene by cells occurs due to the developmental noise from the initial condition, which subsequently results in phenotypic variance in the population.

## Parasite population dynamics

Unlike host phenotype (gene expression levels), which is partially stochastic, the parasites' phenotype is determined uniquely by its genotype. Hence parasites of a given genotype share a common phenotype and parasite population dynamics can be modelled by deterministic equations (instead of individual-based simulations).

The population of each parasite $P_i$ increases in proportion to the host population with the corresponding phenotype, where the host density $H_i$ to be attacked by the parasite strain $i$ is determined by the host population that satisfies $s(x_j) = i_j$. Further, there is a mutation from other parasite types $P'_j s$. Because parasite types are represented by a binary string of length $l_p$, they are represented in a $l_p$-dimensional hypercube space. Each distinct genotype $\upsilon_j = (i_1, \ldots, i_{l_p})$ is represented by a vertex on the cube. Mutations in the parasite genotypes occur by flipping each gene $i_l = 0 \leftrightarrow i_l = 1$ at the mutation rate $D_p$. This is represented by diffusion in the hypercube (e.g., $010 \leftrightarrow 000$) determined by a diffusion constant $D_p$. Therefore, the population dynamics is expressed as

$$\frac{dP_i}{dt} = c/T_p(H_i - \bar{H})P_i + D_p \sum_{j=1}^{N_p} \kappa_{ij}(P_j - P_i) \tag{6}$$

where $\kappa_{ij}$ takes unity only if the binary string of $j$ can change to that of $i$ via a single mutation, otherwise it equals zero. We assume competition within all the parasite genotypes, such that the total number of populations is assumed to be fixed, to avoid the divergence of parasite populations. Accordingly, we normalize the density $P_i$ such that $\sum_{i=1}^{2^{l_p}} P_i = 1$. Thus, the reduction term $\bar{H} = \frac{\sum H_i}{2^{l_p}}$ is introduced ensuring that $\sum_{i=1}^{2^{l_p}} \frac{dP_i}{dt} = 0$ and $\sum_{i=1}^{2^{l_p}} P_i = 1$ remain satisfied. We computed the population dynamics of parasites with a given generation time $T_p$ (s). We set the initial density $P_i$ to take the same value for each type, i.e., $P_i = \frac{1}{2^{l_p}}$ (The result below, however, is independent of this initial distribution). The generation time of parasites is shorter than that of hosts. Therefore, the fittest type would quickly become dominant.

## Phenotypic variances attributed to noise and mutation

The model includes a noise component that allows gene expression level $x_i^{(l)}$ to vary even among individuals sharing the same gene regulatory network. We define the phenotypic variance $V_p^{(l)}(i)$ as the variance of the phenotype $x_i^{(l)}$ over the entire host population with different genotypes. Here, the variation of $x_i^{(l)}$ to give $V_p^{(l)}(i)$ arises both from noise and from network mutations. In addition, we compute the isogenic phenotypic variance $V_{ip}^{(l)}(i)$ of the gene expression level $x_i^{(l)}$ within each host genotype $l$. Because of the noise term $\sigma\eta_i$ in Eq (1), the expression level $x_i$ for the same host genotype (network) varies between individuals and this variance is positive. We define two phenotypic variances to distinguish between these sources: $V_{ip}(i)$ and $V_g(i)$[23, 25]. The variance $V_{ip}^{(l)}(i)$ is defined as the variance of gene expression levels in an isogenic population, i.e., the phenotypic variance of $x_i$ caused by noise in the gene expression dynamics within the clonal population of hosts, i.e., sharing the common $J_{ij}$. In contrast, $V_g^{(l)}(i)$ is defined as the phenotypic variance due to genetic mutation, as computed by the variance of $\bar{x}_i^{(l)}$ within a population of initial genotype $i$ in which each individual receives one random mutation, where $\bar{x}_i^{(l)}$ is the average over the clonal distribution of host $l$. Details describing the calculation of variances $V_{ip}$ and $V_g$ are provided in the *Methods* section. Here, as we are interested in the variance of the expression of output genes, we term the average variances of the output genes ($i = 1, 2, \ldots, 8$) as $V_{ip}$, $V_g$, and $V_p$, thereby omitting the $l$ notation. In addition, we define the average variances of the target genes corresponding to parasite attacks as $V_{ip}^t$ and $V_g^t$.

Note that developmental noise (the $\sigma\eta$ term in Eq (1)) has no significant effect on host fitness once gene expression dynamics have reached a stable state with all $x_i$ close to 0 or 1. This is because host growth depends only on mean gene expression levels and parasite attack depends on rounded values (whether $x_i$ is larger or smaller than 0.5). However, noise is important, because when amplified in the GRN, it may cause the initial dynamics to converge to a different stable state (i.e., noise-induced switches in gene expression). Regarding phenotypic variances, we discuss two types of robustness: mutational robustness and developmental robustness. Mutational robustness describes the "ability" of a GNR to prevent changes in gene-expression patterns due to mutations in the network structure, and developmental robustness describes the ability to prevent changes in expression patterns due to noise-induced fluctuations in gene expression level during the initial network dynamics.

# Results

## Parasite interaction accelerates host diversity

We started simulations with $N$ host cells of an initial gene-regulatory network created by creating random paths with probability $\alpha = 0.3$. We took 50 replicated populations to obtain the statistics. The initial condition for each cell is given by a state in which only the input genes are expressed: all other genes are initialized with expression levels drawn uniformly between 0 and 0.05. This same initial condition was also applied to the daughter cells after each cell division. In the following results, gene regulatory networks (GRNs) that went extinct as a result of parasite interactions are excluded immediately from the sample at the initial time $t = 0$.

Fig 1 shows the population dynamics and fitness values of the host in three parasite environments ($c = 0$, 2, and 4) in the absence of noise ($\sigma = 0$). In the absence of parasite interaction, i.e., at $c = 0$, the host population is dominated by GRNs that generate a phenotype expressing all target genes, such as type "111" (black line), which dominate the host population (see Fig 1A). Then, if sufficient interactions with the parasites occur, the host population evolves into

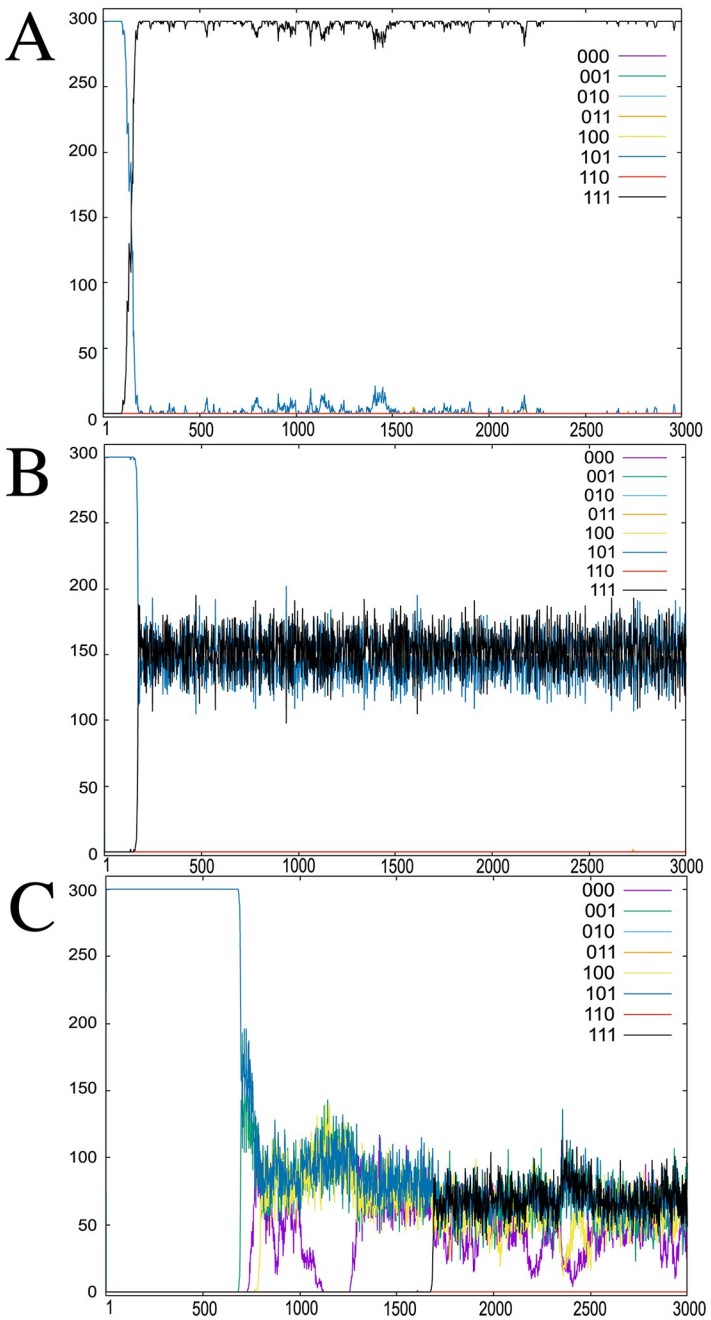

**Fig 1. Dynamics of host phenotypes under interaction with parasites.** The populations of hosts $H_i$ corresponding to the parasite type $(i_1, i_2, \ldots, i_{N_p})$ are plotted against the generation time. The number of parasite types is $N_p = 2^3 = 8$, which corresponds to the on/off expression of the three target genes of the host. The population of each host type $i$ for $H_i$ represents $\{s(x_1), s(x_2), s(x_3)\} = $ "000," "001,". . ., and "111" is indicated by different colors. The interaction strength is $c = 0$ (A), 2 (B), and 4 (C). The noise level is $\sigma = 0.0$. In the absence of parasites ($c = 0$), the population is dominated by the "111" type in which all output gene expression levels are turned on, whereas other phenotypic types with low growth rates ("110," "101," and "011") also coexist at $c = 2$ and 4. In addition, $N = 300$, $M = 64$, $l_{\text{inp}} = 8$, $l_{\text{out}} = 8$, and $l_p = 3$ (see Table 1).

multiple groups with different phenotypes (Fig 1B and 1C). This is explained by the selection pressure to avoid parasite attacks. When the host population is concentrated on the fittest type "11111111," which includes the target gene expressions "111," the parasite population is also concentrated on the corresponding type matching the target "111," which results in suppression of the population of the fittest host types by the interaction with the parasite. Consequently, other host types with less preferred/less fit target patterns have more chances to survive. In short, parasites induce negative frequency-dependent selection pressure, and the effective fitness (Eq (4)) varies over time because of the change in the distribution of parasite genotypes. Thus, multiple phenotypes and genotypes can coexist within a host population. Note that in the absence of noise, the expression dynamics of isogenic individuals always converge to the same pattern.

Next, we computed the phenotypic variance $V_p$ that is the variance of output gene expression levels over the host population with polymorphic genotypes (i.e., network $J_{ij}$) and plotted it as a function of the interaction strength $c$ to determine the extent to which host–parasite interactions enhance phenotypic diversity (Fig 2). Independent of the level of developmental noise, Fig 2 shows a marked increase in the phenotypic variance at a threshold interaction strength $c = c_t \approx 1$. This diversification indicates that multiple phenotypes coexist within the population, as shown in Fig 1B and 1C. In contrast, below the transition point $c_t$, the variance decreases and the host population is concentrated on the fittest phenotype, as shown in Fig 1A.

This critical interaction strength $c_t \approx 1$ follows from the fitness function in Eq (4). The host growth rate is maximized when the expression level of all output genes is close to $x_j = 1$. If the parasite population is concentrated on the type that attacks this fittest phenotype, the population growth is reduced to $cP_m \sim c$, with $P_m$ as the population fraction of the parasite type $m(= 1, \ldots, l_p)$. To allow the phenotype to escape from the parasite attack, one target gene

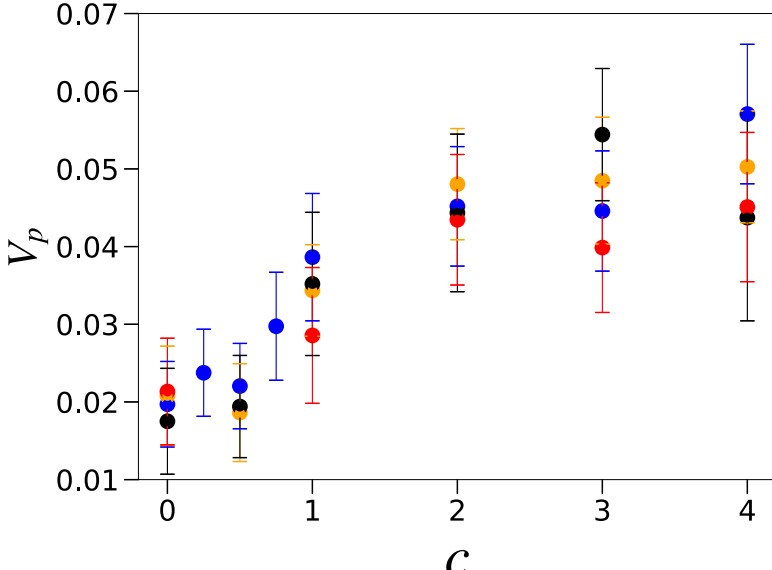

**Fig 2. Total phenotypic variance $V_p$ of the output genes against the parasite interaction strength.** The variance $V_p$ of the output genes plotted against the parasite interaction strength $c$ (bars represent standard deviation). $V_p$ is computed from the expression levels of the output genes over the host population ($N$ = 300 individuals) and plotted as the average $V_p$ over 2500–3000 generations. Each color represents a different noise strength: $\sigma$ = 0.03 (blue), 0.02 (orange), 0.01 (black), and 0.001 (red).

should be switched off, which reduces the raw growth rate by one unit. Then, the raw fitness loss needs to be smaller than the gain by the escape $c$. Since, $P_m \leq 1$, the total fitness gain is positive if $c \gtrsim 1$. Indeed, Fig 2 shows that, for $c > c_t$, the variance increases substantially. When the interaction is strong (e.g., $c = 4$), the number of coexisting host genotypes approaches the maximal number $2^{l_p} = 8$, which corresponds to the total number of parasite types. (see Fig 1C).

In our model, phenotypic variance can have two different origins: genetic variation and noise-induced phenotypic variation. These correspond to two strategies to sustain phenotypic diversity. We investigated how the choice of each of the two adaptive strategies depends on $c$ and $\sigma$, by computing the noise-induced isogenic phenotypic variation, $V_{ip}$, and the variance induced by mutations, $V_g$.

## Evolution of robustness and phenotypic fluctuation

Fig 3 shows evolutionary time courses for the variance components $V_{ip}$ and $V_g$ (for both output and target genes) for various combinations of $c$ and $\sigma$. The temporal averages of $V_{ip}$ and $V_g$ after evolution of the system reaches a stable state are shown in Figs 4 and 5, respectively. These results are independent of the choice of the initial network. Initial values of $V_g$ are high because random networks are not canalized and are highly sensitive to mutation. In addition to the critical interaction strength $c_t$, we can see (e.g., from Figs 4 and 5 against $c$ and $\sigma$) the existence of a critical noise level $\sigma_c \approx 0.02$, above which $V_{ip}$ increases markedly. Based on these two thresholds, behaviors of host gene expression states can be classified into four regimes:

**Regime I: Noninteracting, nonrobust ($\sigma < \sigma_c \sim 0.02$ and $c < c_t = 1$)**

When the noise is below the threshold $\sigma_c = 0.02$ and the interaction is weak ($c < c_t = 1$), $V_{ip}$ is low, whereas $V_g$ is maintained at a high level (Fig 4). The phenotype varies considerably by mutation, which indicates that mutational robustness does not evolve. (A large $V_g$ implies that the phenotypes are not particularly robust against mutations.) Host phenotypes (i.e., gene expression) fluctuate around the fittest type, as shown in Fig 1A, where most of the target genes are expressed, whereas the variances are sustained at moderate values because of genetic variation in the network structure.

**Regime II: Noninteracting, robust ($\sigma > \sigma_c$ and $c < c_t$)**

The evolution of developmental robustness against the noise in gene expression dynamics occurs when the noise is above a threshold of $\sigma_c = 0.02$ and the interaction is weak ($c < c_t = 1$). In this region, the strength of the interaction is weaker than at the transition point $c_t = 1$, which means that the attack by the parasite is not sufficient to cause phenotypic diversification (Fig 2). Fig 4 shows that the values of both $V_{ip}$ and $V_g$ are low relative to the case with $\sigma < \sigma_c$ and $c < c_t$. The phenotype is not significantly changed by genetic variation. $V_{ip}$ and $V_g$ both decrease while satisfying the inequality $V_{ip} > V_g$ (see Fig A in S1 Appendix), and this leads to the evolution of robustness to both noise and mutation. This supports previous studies [17, 23, 25]. The host population loses both genetic and phenotypic inhomogeneity, which indicates that the output genes are all switched on and exhibit minimal variation with respect to the fittest type.

**Regime III: Interaction-induced genetic diversification ($\sigma < \sigma_c \sim 0.02$ and $c > c_t$)**

The evolution of genetic diversification occurs when the noise level is below the threshold $\sigma < \sigma_c$ and the interaction strength exceeds the transition point $c_t$. $V_g$ remains at high levels, whereas $V_{ip}$ remains small, as shown in Fig 5. Phenotypic diversity is generated by genetic diversity. For each given genotype, the generated phenotype is unique and exhibits almost

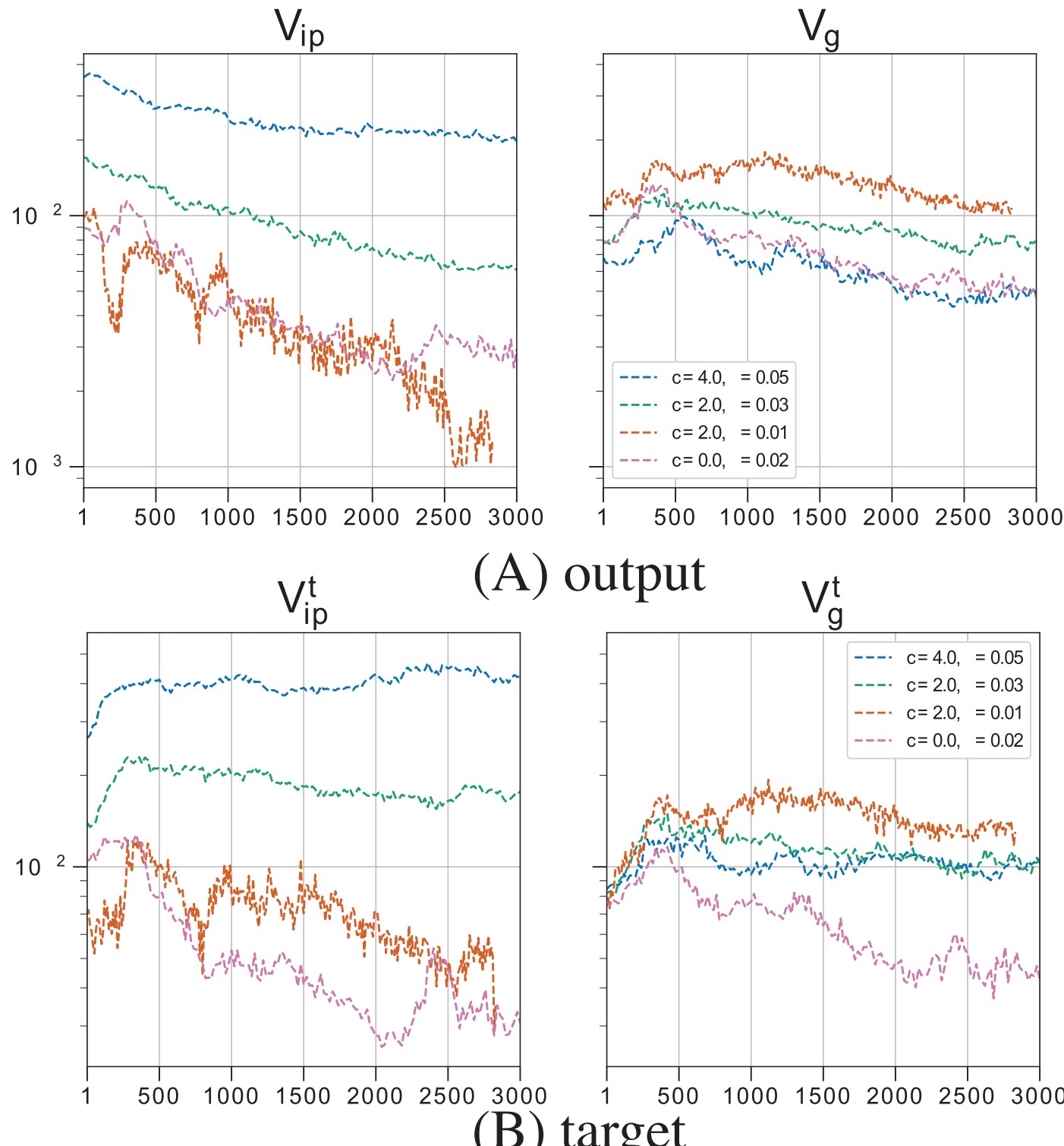

**Fig 3. Evolutionary time course of the average phenotypic variations $V_{ip}$ and $V_g$.** The time course of the average phenotypic variations $V_{ip}$ and $V_g$ (i.e., $V_{ip}^t$ and $V_g^t$). The plot is from the first generation rather than 0th generation. Note that host types that went extinct during the simulation have been excluded. This explains the different initial variance levels between parameter combinations. (A) Average $V_{ip}$ and $V_g$ over all output genes are plotted against the generation for different values of the noise level $\sigma$ and interaction strength $c$, as indicated by different colors. Red: Example of evolution with no host–parasite interactions. Hosts are occupied by individuals with the fittest phenotype, thereby losing phenotypic diversity ($\sigma = 0.02$). Both the isogenic phenotype variation $V_{ip}$ and the genetic variance $V_g$ decrease. Orange: Case with $c = 2$ and $\sigma = 0.01$, which shows that $V_{ip}$ decreases, whereas $V_g$ maintains high values, implying the evolution of genotypic diversity. Green: $c = 2$ and $\sigma = 0.03$. Both $V_{ip}$ and $V_g$ increase. Blue: $c = 4$ and $\sigma = 0.05$. $V_{ip}$ decreases slightly, whereas $V_g$ maintains low values. (B) $V_{ip}^t$ and $V_g^t$ for the target genes. The line colors represent the equivalent conditions as in (A).

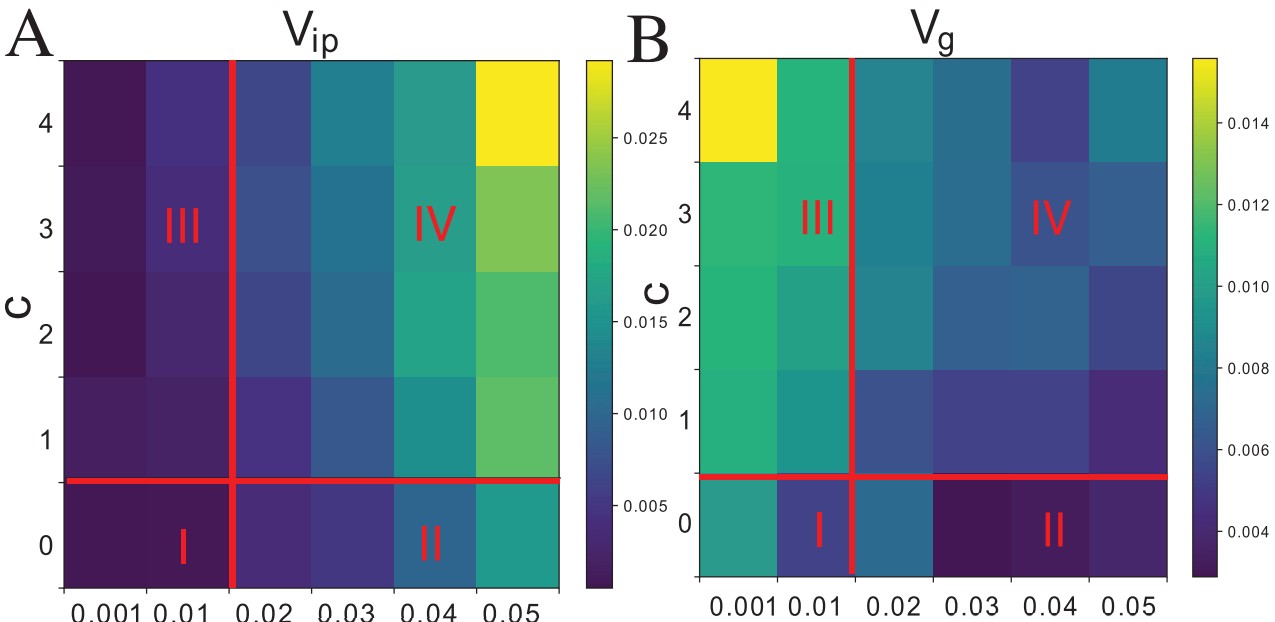

**Fig 4. Dependencies of $V_{ip}$ and $V_g$ on the noise level $\sigma$ and interaction strength $c$.** $V_{ip}$ and $V_g$ for all output genes upon noise level $\sigma$ (horizontal axis) and interaction strength $c$ (vertical axis). The variance values for each parameter are displayed using color maps. The variance is computed by taking the average of 2500–3000 generations. The color maps indicate that the host–parasite interaction enhances $V_{ip}$ and $V_g$. Parameter values of $\sigma$ and $c$ for each square are shown at their left and bottom edge (e.g., the bottom-leftmost square is the results for $\sigma = 0.001$ and $c = 0$) Each parameter regime is categorized into one of four regimes based on the values of $V_{ip}$ and $V_g$ (see the text for details).

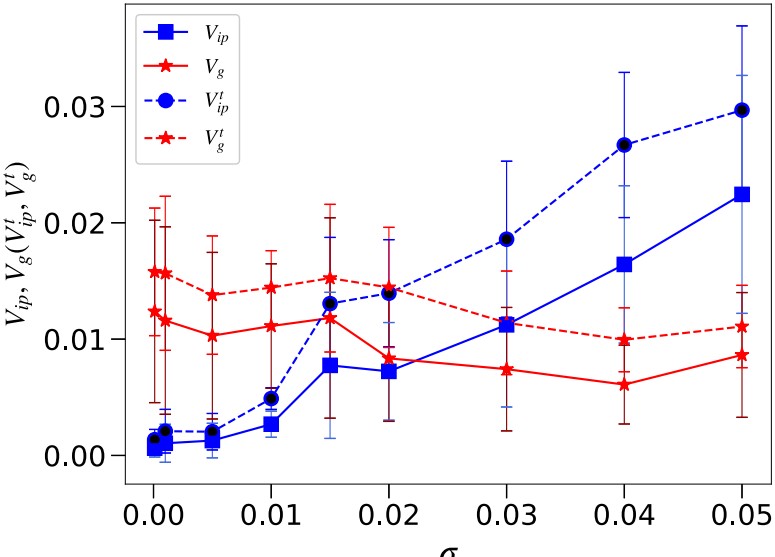

**Fig 5. Transition of $V_g$ and $V_{ip}$ with noise strength.** Dependence of the average variances $V_{ip}$ (red) and $V_g$ (blue) for output genes (solid lines) and $V_{ip}^t$ (red) and $V_g^t$ (blue) for target genes (dotted lines) upon the noise level $\sigma$ for a strong host–parasite interaction ($c = 3$). Bars represent the $\sigma$ standard deviation. $V_{ip}^t$ exceeds $V_g^t$ at $\sigma_c \approx 0.02$. Based on the data in Fig 4, the phenotypic diversification occurs for all parameter values here, but the main origin of the phenotypic variances changes from the mutation to the noise at $\sigma \approx \sigma_c$.

no variation. Multiple groups with different genotypes coexist because of interactions with diversified parasites. The stronger the interaction, the more phenotypes arise, which leads to a larger $V_g$ and higher genetic diversity in the population. For moderate $c > c_t$, the genotypes with the fittest type "111," coexists with genotypes having one target gene switched off ("110," "101," "011," etc.), whereas for $c = 4$, even more host types appear, leading to coexistence, including expression patterns where only one target gene switched on ("100," "010," "001," etc.)

### Regime IV: Interaction-induced phenotypic variation ($\sigma > \sigma_c$ and $c > c_t$)

As exemplified by the case of $c = 3$ and $\sigma = 0.02$, the phenotypic variation of the output gene expression ($V_{ip}$) increases in the initial stage of evolution (up to 500 generations; see Fig 3). Both $V_{ip}$ and $V_g$ maintain high values, while satisfying $V_{ip} > V_g$. Host phenotypes are diversified without resorting to genetic diversification. Here, isogenic hosts can have more than one phenotype. Gene expression levels are diversified by noise, which implies the phenotypic variation of the isogenic population. Variances in expression levels are highest for target genes, whereas those of the other output genes are maintained at high levels (see Fig B in S1 Appendix). Accordingly, the fitness is reduced.

### Transition between Regime III and IV against noise strength

We now explore the transition between the last two regimes. Fig 5 shows the dependence of the variances $V_{ip}$ and $V_g$ of the output genes on the noise level $\sigma$ for $c = 3$. Below the threshold noise level $\sigma_c$, $V_g$ exceeds $V_{ip}$, indicating that there is little robustness against mutation; the diversity of the target gene expression levels is based primarily on genetic variation. In contrast, when the noise level exceeds $\sigma_c$, $V_{ip} > V_g$ and the diversification of gene expression patterns is achieved over isogenic phenotypic variation, i.e., due to noise.

Note that, in Fig 5 $V_g$ decreases slightly with $\sigma$ once $\sigma$ exceeds the critical value $\sigma_c$. This reflects previous results for $c = 0$ (i.e., in the absence of parasites), which showed that mutational robustness decreases as a correlated response to the evolution of developmental robustness against increasing noise levels (Fig A in S1 Appendix). This is because, for a larger noise strength, the acquisition of developmental robustness to noise is needed which also leads to an increase in robustness to genetic mutations. These factors have been investigated in detail in previous studies [23]. When $c$ is nonzero, $V_g$ is larger than the case for $c = 0$, but its decrease against $\sigma$ for the case with $c = 0$ still remains, so that $V_g$ shows slight decrease against $\sigma$.

### Evolution of the noise-induced on/off switches

As mentioned above, once expression levels of host target genes have settled down close to one of the fixed points $x = 0$ or 1, continued variation due to the noise term in Eq (1) has no effect on the interaction with the parasite. The noise is important, however, for the initial network dynamics that determine which of these fixed points is approached in the first place. In order to separate the contributions of noise-induced on/off-switches from those of the noise itself, we define a variance $V'_{ip}$ which only reflects the former. More precisely, $V'_{ip}(i)$ is the variance of the on/off variable $z_i = tanh\beta(x_i - 0.5)$ with $\beta = 100$ and defined $V'_{ip}(i)$ as the variance of $z_i$, computed in the same manner as $V_{ip}(i)$.

Fig 6 shows the dependence of the variances $V'_{ip}$ (output genes) and $V^{t}_{ip}{}'$ (target genes) on the noise level $\sigma$ for $c = 3$. The key difference to Fig 5 is that for $\sigma > 0.02$, $V'_{ip}$ remains essentially constant for both target and output genes, suggesting that the variance due to noise-induced

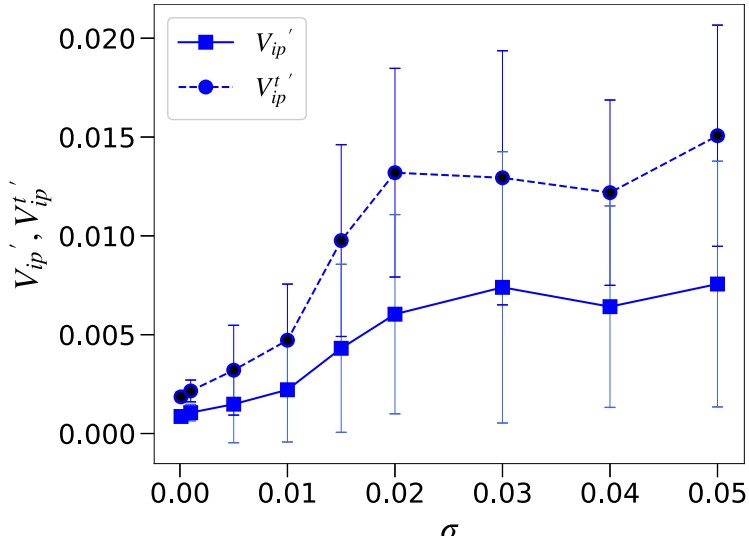

**Fig 6. Transition of $V_{ip}^{t}{}'$ with noise strength.** Dependence of the average variances $V_{ip}'$ for output genes (solid lines) and $V_{ip}^{t}{}'$ for target genes (dotted lines) due to noise-induced switches plotted against the noise level $\sigma$ for a strong host–parasite interaction ($c = 3$). We used the function $z_i = tanh\beta(x_i - 0.5)$ with $\beta = 100$ and defined $V_{ip}'(i)$ as the variance of $z_i$. Bars represent standard deviation.

switches cannot be increased arbitrarily, and that the continued increase of $V_{ip}$ seen in Fig 5 is merely a reflection of the increase in white noise. The reason that $V_{ip}^{t}{}'$ does not increase much beyond $\sigma > 0.02$ will be because the level of phenotypic variation reached by the noise, in conjunction with genetic variation, is sufficient to produce the maximal phenotypic variation. Indeed, $V_p$ with $c = 3$ in Fig 2 is the same regardless of the noise strength. Evolutionary time courses for both $V_{ip}$ and $V_{ip}'$ are shown in Fig E in S1 Appendix. For target genes, both measures increase from their initial value and then stay at a high level, with no evidence for evolution of developmental robustness. For non-target output genes, in contrast, $V_{ip}$ and $V_{ip}'$ decrease over time relative to the values in the initial random networks. This shows that the decrease in the variance has not been observed in the target gene implying that the developmental robustness has not evolved. Also, as discussed in the previous section, $V_g$ decreases with $\sigma$, but not more than the case with $c = 0$, suggesting that there exists some coupling between the $V_{ip}^{t}{}'$ and $V_g^t$. On the other hand, the variance of the non-target output gene ($V_{ip}'$) decreases compared to the initial network, and the developmental robustness evolves in correlation with mutational robustness ($V_g$). At around the noise level ($\sigma = \sigma_c \sim 0.02$), the variances $V_{ip}'$ and $V_{ip}^{t}{}'$ show remarkable increase. Albeit less significant as compared to $V_{ip}^{t}{}'$, $V_{ip}'$ also increases. Although only the on/off of the target gene is relevant to protect against the parasite attack, the noise-induced switch is increased for other genes, due to the coupling in GRN. In the figure, $V_{ip}^{t}{}'$ is of the same level beyond $\sigma = 0.02$. The required switching frequency is of the same level, once at $\sigma \sim \sigma_c$ it occurs. (It increases with the interaction strength $c$, though).

## Dependencies of the variances on genetic change and noise

In the absence of parasites, the variances $V_{ip}$ and $V_g$ decrease as the host phenotype adapts to the environment. This indicates the evolution of robustness to mutation and noise (regime II). Under the pressure of the host–parasite interactions, the gene expression dynamics evolved to

diversify the phenotypes in the manner described above. Here, we discuss the increase in $V_{ip}$ and $V_g$ through evolution.

First, we plot the evolutionary time course of $V_{ip}$ and $V_g$ starting from the initial random networks $J_{ij}$ setting with parameters $c = 3$, $\sigma = 0.03$ in Fig 7A. The variances $V_{ip}^t$ and $V_g^t$ of target genes both increase rapidly because of interactions with parasites. (GRNs that went extinct due to parasite infections of the 0th generation are excluded. Therefore, the variance $V_{ip}$ tends to be larger at $c = 3$ initially.) As shown in Fig 7A(b), the two variance terms exhibit considerably closer correlation between $V_{ip}$ and $V_g$ for the expression of the target gene. Moreover, both variances maintain large values, and the host can escape parasite attacks by producing a different phenotype via mutation. In contrast, Fig 7A(a) shows that the variances of other output genes decreased and developmental robustness increased for those phenotypes that were not attacked by parasites. This decrease, however, is much smaller than that observed in the absence of the host–parasite interactions (see Fig F in S1 Appendix). The decrease implies that the expression of target and non-target output genes is partially decoupled, but the reduction of the decrease suggests some coupling. The evolution of robustness and phenotypic fluctuation occurs in the same GRN's.

## Coping with parasites by phenotypic variation

Next, we investigated whether high variability in phenotype (i.e., phenotypic fluctuation) can evolve in response to the introduction of parasites, even after developmental robustness has evolved. To investigate this issue, we first set $c = 0$ and let the system evolve, and once the host had adapted to the environment and acquired robustness (i.e., after $V_{ip}$ and $V_g$ had decreased from their high initial values), we switched on the interaction by increasing $c$ from 0 to 3. The time course of the variance is shown in Fig 7B. After switching the interaction strength, the host growth rate decreases temporarily before increase in phenotypic variation evolves (see Fig G in S1 Appendix). Then, both the variances $V_{ip}^t$ and $V_g^t$ increase. Thus, notwithstanding the prior evolution of developmental robustness, phenotypic fluctuations can evolve in response to parasite infection. The phenotypic fluctuations ($V_{ip}$) in the host population show a notable increase within 2000 generations. In addition to the target genes, the fluctuations in the expression level of the output genes also increase. Hence, the output and target genes are not completely decoupled.

A comparison of Fig 7A and 7B shows that the final variances are nearly identical. This means that the variances in both target and output genes after evolution are independent of the initial conditions. To increase the phenotypic variance of the target genes, variances of other gene expressions need to be maintained to a certain degree because of gene interactions through the GRN. This trend of a weak increase in the phenotypic variance for all genes holds for $c > c_t$ (see Figs C and D in S1 Appendix).

## Trade-off between growth and tolerance

We examined how the increase in the phenotypic variances caused by parasite interactions influences the growth rate of a single host. Parasites concentrate their attacks on the hosts with the most frequent phenotype in the population. Fig 8 plots the relationship between $V_{ip}$ and the raw fitness, and it shows the trade-off between the two. Individuals exhibiting high plasticity tend to have lower growth rates because of the uncertainty in the expression of output genes; however, they are attacked less by the parasites. As the variance $V_{ip}$ is larger, the parasite attacks are reduced. Consequently, the host genotypes with larger $V_{ip}$ have a larger chance of survival, even if their growth rate as a single cell is lower. In contrast, in region III with large $V_g$, host populations are adapted to parasite infection through genetic variation. In this case,

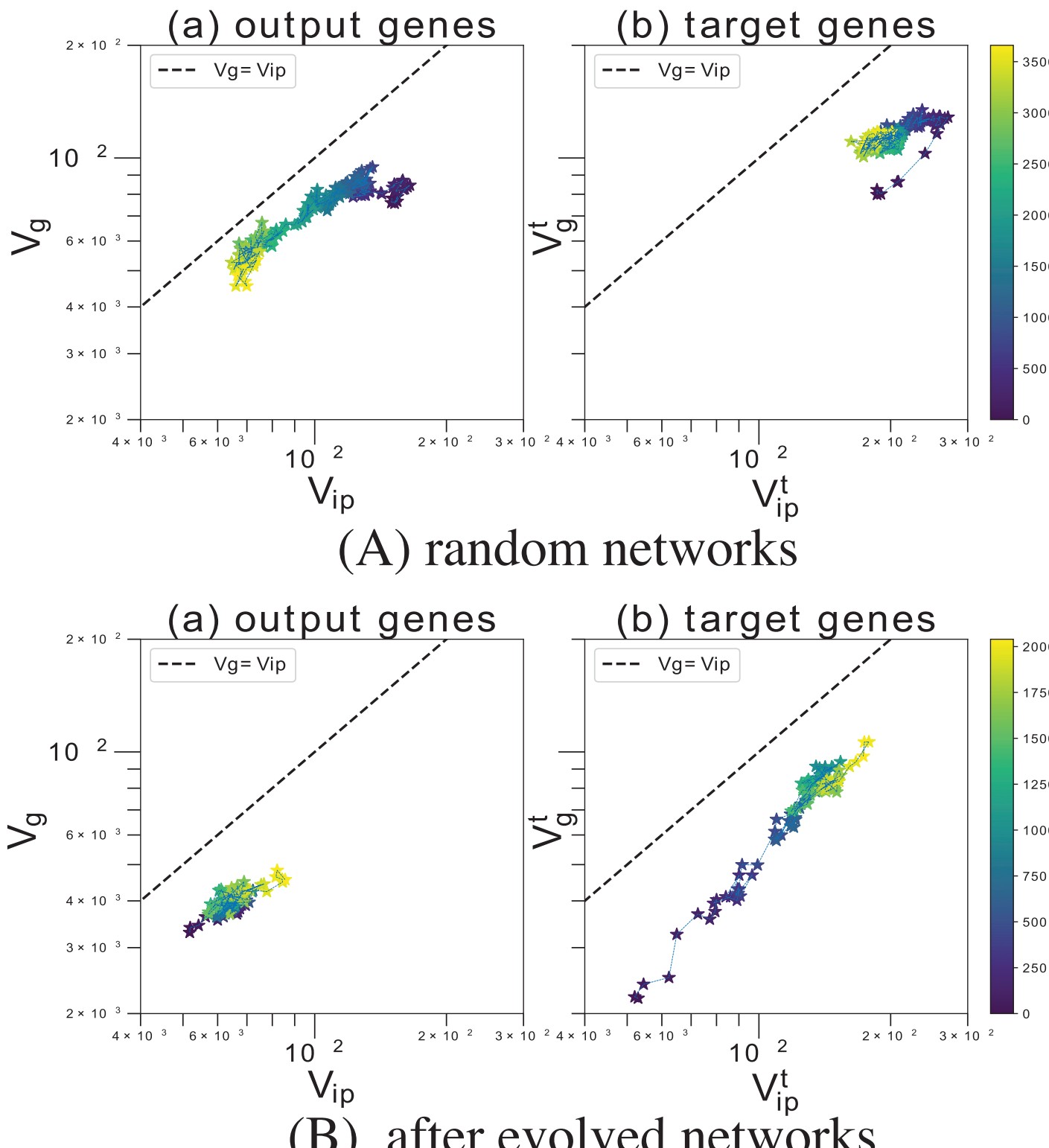

**Fig 7. Time course of the variances ($V_{ip}$, $V_g$) after switching interaction strength.** (A) The time course of the variances ($V_{ip}$, $V_g$) for output genes (a) and the variance ($V_{ip}^t$, $V_g^t$) for target genes (b) plotted over generations for $c = 3$ when starting with initial random networks (circle; dotted lines). The variances are computed from the isogenic variance over 100 iterations. The time course over generations is plotted for the variances of gene expression. The plots cover 3500 generations. We set the noise level to $\sigma = 0.03 > \sigma_c$. The two variances of the non-target output genes decrease, while $V_g < V_{ip}$ is maintained throughout the evolutionary course (a). Conversely, under

host–parasite interactions, $V_{ip}^t$ and $V_g^t$ show a correlated increase (b). (a) shows that the variances of other output genes not attacked by parasites decreased. However, this decrease is much smaller than that corresponding to the absence of host–parasite interactions. (B) The time course of the variances ($V_{ip}^t$, $V_g^t$) for target genes (a) and variance ($V_{ip}$, $V_g$) for output genes (b) over generations for $\sigma = 0.03$ when switching interactions $c$ from 0 to 3 (asterisks; bold lines). For the first 2500 generations, the evolution of the host was modeled without parasites ($c = 0$). Up to this generation, the evolution of developmental and mutational robustness was complete $V_{ip}$ and $V_g$ decreases. Then, we introduced the interaction with the parasites by changing $c$ from 0 to 3. The variances were computed from the isogenic variance over 20 iterations. The time course of ($V_{ip}$, $V_g$) and ($V_{ip}^t$, $V_g^t$) over generations after this switch is plotted for the output and target genes. The color bar indicates the number of generations since the switch. The variances $V_{ip}^t$ and $V_{ip}$ both increase because of interactions with the parasites, which results in the evolution of phenotypic variation. Although this increase is prominent for target genes (b), those for other output genes also show a slight increase up to 2000 generations (a). Variances $V_{ip}^t$ and $V_g^t$ finally reach the same values as those in (A). Other model parameters: $N = 300$, $M = 64$, $l_{\text{inp}} = 8$, $l_{\text{out}} = 8$, and $l_p = 3$.

low phenotypic fluctuations and high raw fitness are compatible. Here, target genes and non-target output genes are well decoupled and raw fitness tends not to decrease even if the expression of target genes is variable (Fig 8 $\sigma = 0.01$ and $\sigma = 0.001$ case). Therefore, the fitness costs are higher for the variance by noise ($V_{ip}$) than that by the genetic variation ($V_g$). The lack of fluctuations, however, makes the host population more susceptible to parasite infections when genetic variation is suddenly reduced because of extinction.

## Discussion

In this study, we investigated the evolution of phenotypic variances using host gene expression dynamics within a regulatory network in the presence of a host–parasite interaction. If the interaction is weak, the host with the fittest phenotype evolves to reduce phenotypic variances. In contrast, if the interaction is sufficiently strong, the phenotypic variance of the host increases as strains evolveadaptation to decrease attack by specific parasites. We identified two strategies—either to increase the noise-induced phenotypic variation ($V_{ip}$) or to increase the genetic variance ($V_g$)—depending on the strength of the noise in the stochastic gene expression. If the noise strength is below the threshold, the diversification is primarily genetic in origin, whereas above the threshold noise-induced phenotypic variation dominates, leading to

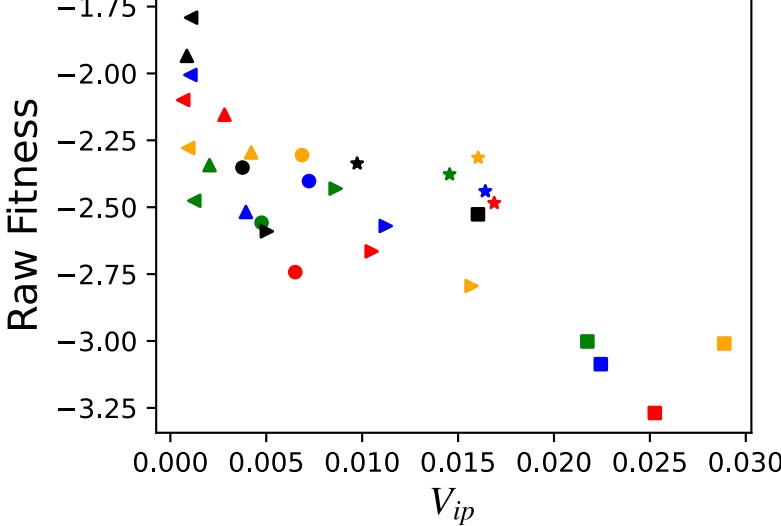

**Fig 8. Trade-off between phenotypic variation ($V_{ip}$) and fitness.** Trade-off between isogenic phenotypic variation ($V_{ip}$) of the output genes and raw fitness $\sum_{i=1}^{l_{\text{out}}} x_i$. Each value is computed by the average from 2500–3000 generations. Each color represents a different interaction strength: $c = 0$ (black), $c = 1$ (green), $c = 2$ (red), $c = 3$ (blue), and $c = 4$ (yellow). Each marker shape represents a different noise strength: $\sigma = 0.05$(◇), $\sigma = 0.04$(⋆), $\sigma = 0.03$(▷), $\sigma = 0.02$(○), $\sigma = 0.01$(△), and $\sigma = 0.001$(◁).

the increase in variances $V_{ip}$ and $V_g$. In the latter case, both variances increase in correlation, and thus, which enhances phenotypic variation and helps avoid parasite attacks.

In a fixed environment without inter-species interactions, both variances $V_{ip}$ (due to noise) and $V_g$ (due to mutation) tend to decrease, which causes loss of phenotypic variation and evolvability. Under host–parasite interactions, the GRN evolves to increase these variances, even after robustness has evolved and the variances have diminished. We classified the diversification of phenotypes into four regimes based on the interaction strength $c$ and noise level $\sigma$. The regimes were classified according to the respective degrees of phenotypic variance caused by phenotypic noise ($V_{ip}$) and genetic variation ($V_g$).

It should be of interest to investigate how the structure of evolved GRN depends on each regime. For example, we examined the distribution of 3-genes motifs [33] for each regime. We detected the increase of motifs that activate target genes in all regimes; however, so far we could not detect a clear difference in the motif distribution among the four regimes yet. The characterization of network structure inherent to each regime remains as a future problem.

Whether biological populations deal with environmental changes by genetic variation or phenotypic variation has been discussed both theoretically and experimentally [3, 34–36]. Indeed, the relevance of species–species interactions to phenotypic variation and genetic diversification has been discussed extensively [37–39]. Therefore, it is interesting to examine how the two strategies for phenotypic diversification studied here are adopted therein.

Here, we observed a trade-off between phenotypic variation and raw fitness. Hosts that evolved the increase in phenotypic variation to deal with parasite interactions tended to exhibit a decrease in raw fitness (i.e., the growth rate). Interestingly, such a trade-off has been observed for predator-induced phenotypic plasticity [39–42] and in bacteria-phage experiments [43], where the bacteria gain resistance to the phage by reducing the competition for resources. Furthermore, it has been suggested that excessively strong phenotypic plasticity may reduce adaptability to climate change [44].

In contrast, host populations that exhibit phenotypic diversification by increasing genetic variation ($V_g$) to reduce the rate of infection do not show a significant decrease in fitness. However, in this case, their phenotypes are less robust against noise or mutation.

The generation of diverse phenotypes against uncertain environmental changes is known as a "bet-hedging strategy" [45–51]. The question of whether to cope with environmental changes by genetic evolution or by phenotypic plasticity has been discussed in relation to the speed of environmental change. The diversification strategies originating from either $V_{ip}$ or $V_g$, as discussed here will provide the basis for the increase in phenotypic variance, which is required for "bet-hedging."

Moreover, the time scale of environmental change is expected to be an important factor in determining which adaptation strategy is selected. Nevertheless, in the present model, the hosts do not receive parasite information explicitly. The effect of parasite infection on the host is considered only as a negative effect on population growth. The influence of the parasite does not have any effect on the host gene expression dynamics. The characteristics of gene expression dynamics are determined by the noise level $\sigma$, which indicates that the transition point $\sigma_c$ is independent of the time scale of the parasite population dynamics (see Fig H in S1 Appendix). Furthermore, because the population density is fixed, the interaction strength is constant in time, which means that the hosts do not become extinct even if they cannot cope with rapid changes in parasite dynamics. It would be interesting to study the evolution of a model in which the host receives the influence of the parasite directly as the input to phenotypic dynamics. For example, in the perceptron model that produces a favorable phenotype for an input, different strategies emerge depending on the environment [50]. By introducing such parasite inputs, host adaptation strategies are enriched.

In summary, we demonstrated the evolution of phenotypic variation and robustness under host–parasite interactions based on the change in phenotypic variances. Depending on the strength of the interaction and the noise level in the gene expression dynamics, either phenotypic variation or genetic diversification evolves. A stronger manifestation of genetic diversification results in speciation. In our forthcoming paper, we aim to show how the isogenic phenotypic variation induced by host–parasite interactions will lead to genetic speciation.

## Methods

We define the two phenotypic variances $V_{ip}$ and $V_g$ as follows.

1. Gene expression can take different values even among individuals of the same genotype because of noise during the development process. The variance, denoted as $V_{ip}^{(l)}(i)$, is defined by the variance of $x_i^{(l)}(t)$ for gene $i$ of the $l$-th host genotype in the isogenic population, i.e., among those with the same genotype (network $J_{ij}^{(l)}$).

2. $V_g^{(l)}(i)$ is defined as the phenotypic variance caused by the genetic variation resulting from mutation for hosts of the genotype $l$. We first computed $\bar{x}_i^{(l)}$, i.e., the average of $x_i^{(l)}$ over noise, for each genotype $l$ to distinguish the variance by mutation and that by noise. Then, we computed the variance of $\bar{x}_i^{(l)}$ over a heterogenic population created by adding random one-step mutations to the interaction matrix $J_{ij}^{(l)}$ of this genotype $l$. In the simulation, there are up to $N$ individuals with different genotypes $J_{ij}^{(l)}$, each of which potentially takes a different gene expression pattern. To calculate the $V_{ip}^{(l)}(i)$ and $V_g^{(l)}(i)$, $L$ individuals with different genotypes are selected at random from the population. For each genotype individual (with genotype $l$), we created $N_c$ clones. Then, we carried out the simulation for each clone under noise. We computed the variance in gene expression $x_i^{(l)}(t)$ caused by noise for each gene at $t = 1000$, after gene expression reaches a steady state. This gives rise to the value $V_{ip}^{(l)}(i)$, that is, the noise-induced variance of gene $i$ for genotypes $l$. Besides the variance, the average $\bar{x}_i^{\,l}$ is obtained for each clone $l$. Subsequently, we created genetic variation in $J_{ij}^{(l)}$ by adding a single random mutation to the network. From each of mutated genotype $l'$, we computed the average $\bar{x}_i^{(l')}$ over $N_c$ clones. From $L$ different mutated genotypes, $V_g^{(l)}(i)$ was computed as the variance of the $\bar{x}_i^{(l')}$ over $L$ different mutants. We used $L = 30$ and $N_c = 50$ for all simulations.

## Supporting information

**S1 Appendix. Supplementary figures. Fig A:** Evolutionary change of the variances $V_{ip}^{lt}$ for output genes for $c = 0$. **Fig B:** Average variances of $V_{ip}(i)$ and $V_g(i)$ (non-target output genes $i = 4, 5, \ldots, 8$). **Fig C:** Average variances of $V_{ip}^t$ and $V_g^t$ (target genes $i = 1, 2, 3$). **Fig D:** Average variances of $V_{ip}(i)$ and $V_g(i)$ (other genes $i = 9, 10, \ldots, 64$). **Fig E:** Evolutionary time course of the average phenotypic variations $V_{ip}$ and $V_{ip}'$. **Fig F:** Evolutionary change of the variances for target genes and output genes for $c = 0$ and $c = 3$. **Fig G:** Time course of the variances $V_{ip}^t$ and the growth rate $\mu$ and $\hat{\mu}$. **Fig H:** Timescale of the parasite changes. **Fig I:** Phenotypic variance $V_{ip}$ and $V_g$ of the output genes against the parasite interaction strength.
(PDF)

## Acknowledgments

The authors would like to thank Tetsuhiro Hatakeyama and Nobuto Takeuchi for stimulating discussions on the study.

## Author Contributions

**Conceptualization:** Naoto Nishiura, Kunihiko Kaneko.

**Data curation:** Naoto Nishiura.

**Formal analysis:** Naoto Nishiura.

**Funding acquisition:** Kunihiko Kaneko.

**Investigation:** Naoto Nishiura, Kunihiko Kaneko.

**Methodology:** Naoto Nishiura, Kunihiko Kaneko.

**Project administration:** Naoto Nishiura, Kunihiko Kaneko.

**Software:** Naoto Nishiura.

**Supervision:** Kunihiko Kaneko.

**Visualization:** Naoto Nishiura, Kunihiko Kaneko.

**Writing – original draft:** Naoto Nishiura, Kunihiko Kaneko.

**Writing – review & editing:** Naoto Nishiura, Kunihiko Kaneko.

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
