## [Decision Letter · Decision Letter 0]

22 Mar 2021

Dear Prof. Kaneko,

Thank you very much for submitting your manuscript "Evolution of Phenotypic Plasticity under Host-Parasite Interactions" for consideration at PLOS Computational Biology.

As with all papers reviewed by the journal, your manuscript was reviewed by members of the editorial board and by several independent reviewers. In light of the reviews (below this email), we would like to invite the resubmission of a significantly-revised version that takes into account the reviewers' comments.

We cannot make any decision about publication until we have seen the revised manuscript and your response to the reviewers' comments. Your revised manuscript is also likely to be sent to reviewers for further evaluation.

Sincerely,

Jacopo Grilli

Associate Editor

PLOS Computational Biology

Natalia Komarova

Deputy Editor

PLOS Computational Biology

Reviewer's Responses to Questions

**Comments to the Authors:**

Reviewer #1: See the attachment

Reviewer #2: see attachment

**Have all data underlying the figures and results presented in the manuscript been provided?**

Reviewer #1: None

Reviewer #2: **No: **Computer code and simulation results should be provided.

PLOS authors have the option to publish the peer review history of their article (what does this mean?). If published, this will include your full peer review and any attached files.

Reviewer #1: **Yes: **Anton S. Zadorin

Reviewer #2: No
---

## [Decision Letter · Decision Letter 1]

6 Jul 2021

Dear Prof. Kaneko,

Thank you very much for submitting your manuscript "Evolution of Phenotypic Fluctuation under Host-Parasite Interactions" for consideration at PLOS Computational Biology.

As with all papers reviewed by the journal, your manuscript was reviewed by members of the editorial board and by several independent reviewers. In light of the reviews (below this email), we would like to invite the resubmission of a significantly-revised version that takes into account the reviewers' comments.

We cannot make any decision about publication until we have seen the revised manuscript and your response to the reviewers' comments. Your revised manuscript is also likely to be sent to reviewers for further evaluation.

Sincerely,

Jacopo Grilli

Associate Editor

PLOS Computational Biology

Natalia Komarova

Deputy Editor

PLOS Computational Biology

Reviewer's Responses to Questions

**Comments to the Authors:**

Reviewer #1: The authors have replied all my questions and have resolved all my concerns. I recommend the article for the publication. The reasons for my recommendation are explained in my first review.

Reviewer #2: General comments:

-----------------

I am still not too happy with the revised version, and I think another

major revision is necessary. My main points of critique are the

following:

1) The writing still needs lots of improvement. It might be good if

the authors could have the manuscript checked by a native English

speaker. I make some suggestions below, but they are far from

exhaustive.

2) I think I understand the model assumptions much better now, but

some points need to be made clearer for the reader. I also think that

there is a problem with the measure V_ip, as part of this variance

does not affect fitness (see below). I therefore suggest to look at a

version of V_ip that only looks at on/off switches of genes, not

small variation in expression level.

3) I am still having a hard time understanding some of the results.

While this is in part due to the "black-box nature" of the GNRs, I

think the manuscript might be improved by (i) adding a measure of V_ip

that focuses on on/off switches (see above) and (ii) adding

information about the relation between V_ip and V_g in random

networks.

I detail these comments in the following section.

Major comments:

--------------

- For a single gene, if the function $f$ is assumed to be constant,

the Langevin equation (1) describes an Ornstein-Uhlenbeck process,

in which gene expression levels x_i fluctuate around a mean of f +

epsilon. More precisely, the stationary distribution is Gaussian

with mean f + epsilon \\approx f and variance sigma^2/(2 gamma); this

should be mentioned in the manuscript; for the largest noise level

sigma = 0.05 (and gamma = 0.1), the variance of the OU process equals

0.0125, and the corresponding standard deviation is 0.112, meaning

that, for f close to either 0 or 1, genes can still be essentially

"switched on and off", with expression levels in the vicinity of 1

or 0 (but note that in the latter case, the actual x values can

become negative, unless this is somehow excluded in the simulations;

please clarify). Importantly, the variation around these values is

insignificant for fitness (the interaction with parasites depends

only on "rounded" or discretized values, and growth rate mu depends

on the mean expression level of output genes), even though it is

essential for the dynamics of the gene-regulatory network (i.e.,

for triggering noise-induced variation).

- I think it is useful to introduce the terms mutational robustness

(which prevents changes in gene-expression patterns due to mutations

in the network structure) and developmental robustness

(which prevents on/off switches in the face of the white-noise

fluctuations in gene expression level).

- The variance sigma^2/(2 gamma) of the Ornstein-Uhlenbeck

process (i.e., the variance of small-scale fluctuations in

expression levels) directly contributes to the measure V_ip (which

should be the sum of this variance and the variance due to on/off

switches). This blurs the interpretation of some results, and in

particular, makes it hard to see evidence for evolution of

developmental robustness. For example, in Fig. 4A, V_ip increases

with increasing sigma, but we cannot say how much of this effect is

due to the direct effect of expression-level fluctuations (increase

in sigma^2/(2 gamma)), and how much due to an increase in

noise-induced on/off switches. Similarly, in Fig. 5, the blue lines

(V_ip) beyond sigma = 0.02 appear to be roughly parallel to

sigma^2/(2gamma), which would suggest that noise-induced variation is

triggered around $sigma = sigma_c$, but that nothing much happens

beyond this value in terms of genes being switched on or off.

- Since for the interaction with the parasite only the on/off switches

matter, I suggest to look at a measure V_ip' that reflects only this

variation. Such a measure might be V_ip - sigma/(2gamma), the

variance of x_i after rounding to 0 or 1, or the probability

that a given gene is switched on (note that when x_i values are

discretized to 0 and 1, the resulting variance is the variance of

Bernoulli distribution, which is p(1-p), where p = P(x = 1)). If

this measure goes down (relative to the value in an unevolved

network), it is evidence for evolution of developmental robustness.

- I repeat that V_g is not standing genetic variation (this would be the

variance of $\\bar x$), and in contrast to what the authors claim in

the reply letter, V_g + V_ip does not equal V_p. One way to

see this is that V_ip is expressed within a generation, but V_g only

between generations. So be careful to not call V_g the variance "due

to genetic differentiation". A population might be genetically

homogeneous and still have positive V_g.

- I am still struggling to understand the initial variance values in

Fig. 3 and 6 (e.g., why is V_g so low initially in Fig. 3[i]?). You

write that genotypes that die out due to parasitism are excluded

from the sample, but do these extinctions happen immediately at t =

0?

- In addition, you write that in a previous paper you have shown that

V_ip and V_g are correlated (I suppose this means positively

correlated?). For readers of the present paper, I think it would be

useful to repeat these results, that is, to show the relation of

V_ip (or V_ip') and V_g in random networks for various values of

sigma.

- With the additional information mentioned above, I hope to better

understand some of the results. Currently, I still have the

following questions:

- Fig. 1(i): I don't understand why for c = sigma = 0, there are such

large fluctuations in host phenotype frequencies, with the optimal

phenotype sporadically dropping to about 50% of the total

population. As you notice this is evidence that mutational

robustness does not evolve in regime I, but why? And where do

these fluctuations come from exactly?

- As mentioned above, in Fig. 4A, it would be nice to see the

behavior of V_ip' (variance due to on/off switches only), and in

particular, whether it increases as a function of sigma. If

V_ip' does not increase (or increases less than in random

networks) this would be evidence for developmental robustness.

In addition, we see that V_g decreases as a function of sigma

(mutational robustness). This might be either a correlated

response to increased developmental robustness, or (in case

developmental robustness is absent) and kind of "compensatory"

evolution (but this would mean that evolution of developmental

and mutational robustness can be decoupled, or at least that one

evolves easier than the other).

- In Fig. 5, as also mentioned above, it seems to me that V_ip'

does not increase beyond sigma = 0.02. In this case, why does

V_g decrease? Again, is this a correlated response to the

evolution of developmental robustness (which keeps V_ip

constant), or is there a different cause? For example, I have

been wondering whether V_g automatically decreases with sigma

(maybe mutations have less effect if expression levels

fluctuated all the time anyway?, see my question about random

networks above).

Some more question:

- In the presence of noise-induced variations, do expression patterns

remain for individual cells once they have settled down to a stable

state, or do they occasionally change? In other words, does

noise-induced variation manifest only in the differences between

isogenic individuals, or also in single individuals over time?

- In the absence of noise (sigma = 0), do the expression dynamics of

isogenic individuals always converge to the same pattern, or are

can there be differences cause by differences in the initial random

expression levels?

Specific comments:

-----------------

- Equation (1): As mentioned above, for fixed $f$, this describes an

Ornstein-Uhlenbeck process (a random walk with a tendency to return

to the mean), in which $x_i$ fluctuates around $f + \\epsilon$ with

variance $\\sigma/(2\\gamma)$.

- 93: Make it easier for the reader to understand that (i) there

are are M (= 64) genes in total, (ii) the first $l_{out} = 8$ genes

are output genes affecting growth, (iii) of these the first $l_p =

3$ are target genes, and (iv) the last $l_{inp} = 8$ are input

genes.

- 96: Maybe the formula in this line is needlessly complicated

here, and it is enough to just say that $\\eta$ describes white noise

which uncorrelated across genes. This is especially true if you

state that equation (1) describes an Ornstein-Uhlenbeck process (and

maybe mention that it is a Langevin equation, even though most

biologists will not know this term).

- 94: k should be l_inp.

- 98: Does epsilon play any role at all for the results?

- 102: I = 1 means that input genes are always expressed.

(Indeed, the idea of external input seems unnecessary here; you

might just say that these genes are always/constitutively

expressed).

- 104: output genes have already been mentioned in line 94.

- 106: If I understand correctly, N = 300 is constant throughout,

as in the Moran model.

- Equation (3) and (4): I suppose that the growth rate is evaluated

instantaneously at each moment and can change over time (so it should be

$\\hat \\mu^{(l)}(t)$ in equation (5).

- 130-1: the part about parasite dynamics should be moved to the next

paragraph.

- 136: When there are 300 cells and one divides, the upper limit N

will always be exceeded, so it is more precise to say that whenever

a cell divides, another randomly selected dies.

- 140: What is s? Seconds? Maybe better to just say between 500 and

1000 time units.

- 141: When the gene-regulatory network reaches a steady state, but

there is phenotypic variation due to noise, do individual networks

settle into a stable state (which may vary between individuals), or

are genes occasionally switched on or off in response to

fluctuations in expression levels?

- 149: alpha = 0.3 is not just an initial choice, but a choice that

remains valid throughout the simulation.

- Equation (6): c is the attack parameter from equation (4). Here it

should probably be replaced by 1/T_p.

- 178: As mentioned above, V_ip has two components: One to the

fluctuations in eta, and one due to noise-induced on/off

switches.

- 185: V_g is not the variance due to genetic differences, but the

variance induced by mutations. In particular, unlike V_{ip}, V_g

is not expressed within a generation, but only between generations.

It is really important to make this clear, especially since readers with a

background in quantitative genetics will tend to assume that V_g is

the genetic variance (i.e., the variance in $\\bar x$), which is not

the case.

- Table 1: Is the mutation rate m = 0.0002 (as in the table) or 0.03

(as in line 143)?

- 200: From which sample? The one for t = 0?

- 241: Again, V_g is not the genotypic variance, but the variance

induced by new mutations.

- 247: Initial values of V_g are high because random networks are not

canalized and highly sensitive to mutation.

- 266-7: V_ip is higher, not lower, under large sigma for small c in Fig. 4

(regime 2).

- 268: Where can I see that V_ip and V_g both decrease (the purple

lines in Fig. 2?)

- 286: c = 4?

- 152: Unlike host dynamics, which are stochastic and

individual-based, parasite dynamics are modelled via a deterministic

model of genotype frequencies.

- 177-180: These two sentences seem redundant, and they disrupt the

connection between "both from noise and from network mutations"

(177) and "We define two phenotypic variances ..." (180).

- 235: the number of coexisting host genotypes approaches the maximal

number 2^{lp} = 8, corresponding to the total number of parasite

genotypes.

- 259: are not

- 312: GRNs that became extinct due to parasite infections are excluded

- 317: and robustness increased

- 320: Fig. S6

- 324: whether high variability in phenotype can evolve in response

to...

- 327: once the host had adapted

- 331: before increase phenotypic variation evolves

- 332: Thus, notwithstanding ..., phenotypic fluctuations can evolve

in response to parasite infection

- 338: A comparison of Fig. 6[i] and [ii] shows the final variances

are nearly identical. This means that the evolution variances in

both target and output genes is independent of the initial

conditions.

- 358: the expression of target genes is variable

**Have the authors made all data and (if applicable) computational code underlying the findings in their manuscript fully available?**

Reviewer #1: **No: **I answer "No" to the question of the computational code availability, as it has not been yet published at the date of the review request, but the authors promised to do so on Github in the future.

Reviewer #2: **No: **Authors say they will upload the code to Github upon publication.

PLOS authors have the option to publish the peer review history of their article (what does this mean?). If published, this will include your full peer review and any attached files.

Reviewer #1: **Yes: **Anton S. Zadorin

Reviewer #2: **Yes: **Michael Kopp
---

## [Decision Letter · Decision Letter 2]

7 Oct 2021

Dear Prof. Kaneko,

Thank you very much for submitting your manuscript "Evolution of Phenotypic Fluctuation under Host-Parasite Interactions" for consideration at PLOS Computational Biology. As with all papers reviewed by the journal, your manuscript was reviewed by members of the editorial board and by several independent reviewers. The reviewers appreciated the attention to an important topic. Based on the reviews, we are likely to accept this manuscript for publication, providing that you modify the manuscript according to the review recommendations.

Sincerely,

Jacopo Grilli

Associate Editor

PLOS Computational Biology

Natalia Komarova

Deputy Editor

PLOS Computational Biology

[LINK]

Reviewer's Responses to Questions

**Comments to the Authors:**

Reviewer #1: I have not issues with the current version of the manuscript and I, as before, recommend it for the publication.

Reviewer #2: In this revision, the authors have answered most of my questions. In

particular, the new Fig. is a real improvement, which shows that the

variance due to noise-induced on/off switches does not increase for

noise levels beyond sigma = 0.02. In would be good to discuss why this

is so. One potential explanation is that the attained level of

noise-induced phenotypic variation, in conjunction with genetic

variation, is enough to produce the maximal phenotypic variation with

8 coexisting target-gene expression patterns.

Editorial comments:

------------------

- Abstract, first "blue" sentences: two typos in plasticity

- Abstract, line 9: where one of the most important...

- Abstract, second "blue" part: By using a dynamic model of gene

expression for the host ... evolution of the genotype-phenotype map

and of phenotypic variances

- 8: Maybe: In this context, phenotypes are shaped by dynamic

processes involving several genetically determined variables, as

well as external and internal noise. Hence phenotypes can

vary not only as a consequence of genetic variation, but even

between isogenic individuals [6–10]. The sum of both sources of

phenotypic variation defines the genotype–phenotype mapping [14–16].

- 31: remain

- 46: latter's

- 81: I still think it would be useful for readers to say that, for

constant f, equation 1 defines an Ornstein-Uhlenbeck process, and to

give the equilibrium variance of sigma^2/(2gamma).

- 92: total gene number

- 110: constant at N = 300 throughout all simulations

- 125: Let m be the parasite type

- 142: after around 200 time units

- 145: value

- 145: one of the others

- 151: alpha = 0.3, and alpha remains valid...

- 151: of this choice

- 153: Unlike host phenotype (gene expression levels), which is

partially stochastic, the parasites' phenotype is determined uniquely by

its genotype. Hence parasites of a given genotype share a common

phenotype and parasite population dynamics can be modelled by

deterministic equations (instead of individual-based simulations).

- 192: the variance ... within a population of initial genotype i in

which each individual receives one random mutation.

- 199: Note that developmental noise (the sigma eta term in equation

1) has no significant effect on host fitness once gene expression

dynamics have reached a stable state with all x_i close to 0 or 1.

This is because host growth depends only on mean gene expression

levels and parasite attack depends on rounded values (whether x_i is

< or > 0.5). However, noise is important, because when amplified in

the GRN, it may cause the initial dynamics to converge to a

different stable state (i.e., noise-induced switches in gene

expression). Regaring phenotypic variances, we discuss two types of

robustness: mutational robustness and developmental robustness.

Mutational robustness describes the "ability" the ability of a GNR

to prevent changes in gene-expression patterns due to mutations in

the network structure, and developmental robustness describes the

ability to prevent changes in expression patterns due to white-noise

fluctuations in gene expression level during the initial network

dynamics.

- 209: We started simulations with N host cells of an initial

gene-regulatory network created by creating random paths with

probability alpha = 0.3.

- 210: What are the 50 samples? 50 individuals from a population, or

50 replicated populations?

- 211: The initial condition for each cell is given by a state in

which only the input genes are expresses: all other genes are

initialized with expression levels drawn uniformly between 0 and

0.05. This same initial condition was also applied to the daughter

cells after each cell division.

- 214: that went extinct

- 216: add "in the absence of noise (sigma = 0)" at the end of the

sentence

- 229: distribution of parasite genotypes

- 231: Note that in the absence of noise...

- 233: Next, we computed

- 233: expression levels

- 236: Independent of the level of developmental noise, the Figure

shows a marked increase...

- 238: This sentence is redundant

- 255: These correspond to two strategies

- Fig. 3, new sentence: Note that the plot starts with the first

rather than te 0th generation, and that host types that went extinct

during the simulation have been exluded. This explains the different

initial variance levels between parameter combinations.

- 261: after evolution of the system reaches a stable state

- 263: These results are independent of the choice of the initial

network

- 287: to both noise and mutation

- 294: whereas V_ip remains small

- 299: Maybe: For moderate c > c_t, the genotypes with the fittest

type 111 coexists with genotypes having one target gene switched off

..., whereas for c = 4, even more host types appear, leading to

coexistnence of all eight target gene expression patterns.

- 307: The sentence starting with "In addition" seems redundant with

respect to the previous one.

- 309: Variances in expression levels are highest for target genes

- 320: Note that, in Fig. 5, V_g decreases slightly with sigma once

sigma exceeds the critical value sigma_c. This reflects previous

results for c = 0 (i.e., in the absence of parasites), which showed

that mutational robustness decreases as a correlated response to the

evolution of developmental robustness against increasing noise

levels (Fig. S8).

- 329: As mentioned above, once expression levels of host target genes

have settled down close to one of the fixed points x = 0 or 1,

continued variation due to the white-noise term in Fig. 1 has no

effect on the interaction with the parasite. The noise is important,

however, for the initial network dynamics that determine which of

these fixed points is approached in the first place. In order to

separate the contributions of noise-induced on/off-switches from

those of the noise itself, we define a variance V_ip' which only

reflects the former. More precisely, V_ip'(i) is the variance

of the on/off variable z_i = ..., computed in the same manner as

V_ip(i).

- 336: After the first sentence, first discuss Fig. 6: "The key

difference to Fig. 5 is that for sigma > 0.02, V_ip' remains

essentially constant for both target and output genes, suggesting

that the variance due to noise-induced switches cannot be increased

arbitrarily, and that the continued increase of V_ip seen in Fig. 5

is merely a reflection of the increase in white noise."

- Then go on to discuss Fig. S9: E.g., "Evolutionary time

courses for both V_ip and V_ip' are shown in Fig. S9. For target

genes, both measures increase from their initial value and then stay

at a high level, with no evidence for evolution of developmental

robustness. For non-target output genes, in contrast, V_ip and V_ip'

decrease over time relative to the values in the initial random

networks. This shows that ..."

- 345-351: I don't understand what you are trying to say here.

- 352-355: Thanks for adding this, but it be better placed earlier in

the paper, for example in the model description, after line 151.

- in the 0th generation?

- 368: closer than what?

- 370: remove "the" before developmental robustness

- 379: after developmental robustness has evolved

- 381: and let the system evolve

- Caption Fig. 7: It seems that there is a problem with the sentence

right after the blue insert.

- 418: within a regulatory network

- 421: the phenotypic variance of the host increases as strains evolve

adaptation to decrease attack by specific parasites.

- 427: which enhances phenotypic variation and helps...

- 428: The new sentence seems redundant.

- 430: both variances

**Have the authors made all data and (if applicable) computational code underlying the findings in their manuscript fully available?**

Reviewer #1: **No: **The program text is promised to be published in the future.

Reviewer #2: **No: **Simulation code should be made available

PLOS authors have the option to publish the peer review history of their article (what does this mean?). If published, this will include your full peer review and any attached files.

Reviewer #1: **Yes: **Anton S. Zadorin

Reviewer #2: No

Figure Files:

Data Requirements:

Reproducibility:

References:

---

## [Editor Report · Decision Letter 3]

19 Oct 2021

Dear Prof. Kaneko,

We are pleased to inform you that your manuscript 'Evolution of Phenotypic Fluctuation under Host-Parasite Interactions' has been provisionally accepted for publication in PLOS Computational Biology.

Best regards,

Jacopo Grilli

Associate Editor

PLOS Computational Biology

Natalia Komarova

Deputy Editor

PLOS Computational Biology

---

## [Editor Report · Acceptance letter]

1 Nov 2021

PCOMPBIOL-D-21-00026R3 

Evolution of Phenotypic Fluctuation under Host-Parasite Interactions

Dear Dr Kaneko,

I am pleased to inform you that your manuscript has been formally accepted for publication in PLOS Computational Biology. Your manuscript is now with our production department and you will be notified of the publication date in due course.

With kind regards,

Zsofia Freund
